# Effect of Amino Acids on Folates Accumulation in Wheat Seedlings during Germination under Red Light Radiation

**DOI:** 10.3390/molecules27206868

**Published:** 2022-10-13

**Authors:** Chong Xie, Pei Wang, Jianwei Chang, Qiaoe Wang, Yongbin Han, Runqiang Yang

**Affiliations:** 1College of Food Science and Technology, Whole Grain Food Engineering Research Center, Nanjing Agricultural University, Nanjing 210095, China; 2Beijing Key Lab of Plant Resource Research and Development, Beijing Technology and Business University, Beijing 100048, China

**Keywords:** wheat seedlings, amino acids, folate, enrichment

## Abstract

Deficiency of folates can cause various health problems, and germination is a potential way to enrich folates in grain-based food materials. In the present study, the effects of six amino acids (phenylalanine, tyrosine, tryptophan, glutamate, γ-aminobutyric acid, and *p*-aminobenzoic acid) on folate accumulation during wheat germination under red light radiation were investigated, and an optimized combination of amino acids for promoting folate enrichment was established. The results showed that applying phenylalanine, tyrosine, tryptophan, glutamate, or *p*-aminobenzoic acid to wheat seedlings during germination can significantly increase the content of total folates through activating the synthesis of the precursors for folate synthesis (pterin and *p*-aminobenzoic acid) or condensation of these two moieties. Meanwhile, up-regulation of corresponding genes was observed by measuring their expressions to investigate the mechanism for promoting the accumulation of folates. The highest content of folates (ca. 417 µg/100 g DW) was observed when the germinated wheat was cultured with a mixture of 1.5 mM phenylalanine, 0.5 mM tyrosine, 0.5 mM tryptophan, 0.75 mM *p*-aminobenzoic acid, and 0.5 mM glutamic acid, which was 50% higher than the control seedlings. This study established a promising and practical approach to enhance the accumulation of folates in wheat seedlings.

## 1. Introduction

Folates (also known as vitamin B9) are a group of water-soluble micronutrients which are involved in various essential metabolic processes, including the production of erythrocytes, biosynthesis of DNA, differentiation and proliferation of somatic cells, and consumption of carbohydrates [1]. Plants and microorganisms can synthesize folates de novo, but humans cannot synthesize this nutrient and so must obtain it from food [2]. The recommended daily intake (RDI) of folates is 400 μg for adults or 600 μg for pregnant women, and insufficient intake of folates will result in various health problems, such as megaloblastic anemia and neural tube defects [2]. Unfortunately, it is difficult to reach the RDI of folates from one’s daily diet—especially for pregnant women and the elderly—without supplements. The mandatory fortification of folic acid—the man-made form of folates—in food has been applied in many countries and a significantly lower occurrence of folate-related disorders, such as neural tube defects, spina bifida, and encephalocele, was achieved [3,4]. However, reports of adverse health outcomes from excess intake of folic acid have aroused wide concern regarding this mandatory fortification [5], so developing food enriched with a natural form of folates is required.

As the staple food in many countries, wheat (*Triticum aestivum*) is an ideal food material for enrichment with folates. To increase the content of folates in wheat, many strategies have been developed, such as metabolic engineering [6], grain germination [7], and the use of elicitors [8]. Germination has been proven to be an effective and economical way to increase the level of folates in cereal grains. Our previous study showed that red light radiation can further enhance the accumulation of folates in wheat seedlings during germination [9]. Folates consist of a pterin, a *p*-aminobenzoic acid (*p*ABA), and one or more glutamates [10]. Glutamic acid (Glu) and *p*ABA are precursors in folate biosynthesis, and the addition of these molecules has been shown to enhance folate synthesis in plants [11]. Gamma-aminobutyric acid (GABA) is a non-protein amino acid which plays a versatile role in plants and can mediate the synthesis of many bioactive compounds, such as phenolics, as a signal molecule [12]. Aromatic amino acids, such as tyrosine (Tyr), phenylalanine (Phe), and tryptophan (Try), are potential molecules for enhancing folate accumulation because they are synthesized by chorismite, which is a precursor in *p*ABA biosynthesis, through the shikimate pathway [13].

In the present study, the effects of six amino acids (Glu, *p*ABA, GABA, Phe, Try, and Tyr) on folate accumulation during wheat germination under red light treatment were investigated. Furthermore, an orthogonal experiment was applied to optimize the combination of amino acids for folate accumulation. In addition, the content of precursors and expression of genes related to the biosynthesis of folates in wheat seedlings were investigated to study the mechanism of folate accumulation in germinated wheat under red light treatment with the addition of amino acids.

## 2. Results

### 2.1. Content of Total Folates in Wheat Seedlings after Germination

Without the addition of amino acids, ca. 280 μg/100 g DW of folates were found in wheat seedlings after germination treated with red light (Figure 1). In the seedlings treated with Phe, Tyr, *p*ABA, and Glu (Figure 1A,B,D,E), when the addition levels were increased, the content of folates first increased and then decreased. The application of GABA (up to 6 mM) had no significant (*p* > 0.05) influence on the content of folates (Figure 1F). Compared to the sprout without any addition, the application of Trp (0.5 mM) increased the content of folates in the wheat seedlings by ca. 32% (Figure 1C). However, increasing the Trp addition (0.5–4.0 mM) cannot enhance its promotion of folate accumulation (Figure 1C). The highest accumulation of folates (ca. 398 μg/100 g DW) among all treatments was observed in the seedlings treated with 1 mM Phe. Meanwhile, the lowest content (ca. 195 μg/100 g DW) of folates was observed in the seedlings treated with 6 mM Phe. The highest accumulations of folates in the seedlings treated with different levels of Tyr and *p*ABA were ca. 342 μg/100 g DW and ca. 320 μg/100 g DW, respectively.

### 2.2. Optimization of Amino Acids Addition for Folate Accumulation

To optimize the combination of amino acids for folate accumulation in the wheat seedlings, an orthogonal design was employed, and four levels of five factors (Phe, Tyr, Try, *p*ABA, and Glu) were selected based on the results on the effects of different levels of individual amino acids on folate accumulation (Section 2.1). As shown in Table 1, within the concentration range considered, the degree of effects of the factors on the total folate content was as follows: Phe > Glu > Tyr > *p*ABA > Trp. By comparing different levels of K values, it was estimated that the best amino acid culture solution combination for accumulation of folate was A_4_B_2_C_3_D_4_E_1_, which contained 1.50 mM Phe, 0.25 mM Tyr, 0.50 mM Trp, 0.75 mM *p*ABA, and 0.5 mM Glu.

To verify the validity of the design, the optimized combination of amino acids obtained from the orthogonal experiment was compared with three random combinations to determine their effects on total folate content. The results showed that the folate content in wheat seedlings cultivated with an optimized amino acid culture solution (ca. 417 μg/100 g DW) was significantly (*p* < 0.05) higher than that of random combinations, which ranged from 306–349 μg/100 g DW (Table 2).

### 2.3. Profile of Folates in Wheat Seedlings with Different Treatments

Tetrahydrofolate (THF) and 5-methyltetrahydrofolate (5-CH_3_-THF) were the main types of folates in wheat seedlings (Figure 2). Except *p*ABA, the addition of other amino acids had a significant promoting effect on the accumulation of THF in wheat seedlings, and the content of THF increased by 46.97–62.93% (Figure 2A). The addition of Phe, Tyr, Trp, and *p*ABA significantly (*p* < 0.05) increased the content of 5-CH_3_-THF in wheat seedlings, and the optimized culture solution showed the best promoting effect, which increased the content of 5-CH_3_-THF to ca. 190 μg/100 g DW (Figure 2B). As shown in Figure 2C, all the treatments significantly (*p* < 0.05) increased the content of 5-formyltetrahydrofolate (5-CHO-THF), and the seedlings treated with Phe had the highest content (ca. 77 μg/100 g DW). Compared with the control seedlings, only Glu increased the content of 10-formylfolic acid (10-CHO-FA), while other amino acids, including the optimized culture solution, significantly (*p* < 0.05) reduced the content (Figure 2D).

### 2.4. Endogenous Free Amino Acids

In the control seedlings, the content of total endogenous free amino acids was ca. 9.8 μg/100 g DW (Table 3). The seedlings treated with Phe had the highest content of endogenous free amino acids (ca. 14.3 μg/100 g DW) followed by the seedlings treated with the optimized culture solution (ca. 10.7 μg/100 g DW). Notably, the content of endogenous Phe in the seedlings treated with Phe increased by 5.51 times when compared with the control seedlings. The addition of *p*ABA decreased the content of threonine by 53.99%. The content of asparagine, glutamine, alanine, Tyr, Phe, lysine, arginine, and proline in wheat seedlings cultured with the optimized solution increased significantly (*p* < 0.05), but the threonine content decreased by 35.97%.

### 2.5. The Precursors of Folate Biosynthesis

The contents of the precursors, *p*ABA and pterin, in folate biosynthesis during germination are shown in Figure 3. The content of endogenous *p*ABA in control wheat seedlings was ca. 80 μg/100 g DW (Figure 3A). Treatments of Phe, Trp, *p*ABA, and optimized culture solution increased the *p*ABA content by 4.42, 4.05, 11.91, and 12.45 times, respectively. There was no significant difference of endogenous *p*ABA content between seedlings treated with *p*ABA and optimized culture solution. The addition of Tyr and Glu had no significant (*p* > 0.05) effect on the content of endogenous *p*ABA.

The content of endogenous pterin in the control wheat seedlings was ca. 36 μg/100 g DW (Figure 3B). The treatments of Phe, Tyr, Trp, and *p*ABA significantly (*p* < 0.05) increased the content of endogenous pterin in wheat seedlings by 63.60%, 26.53%, 25.90%, and 24.88%, respectively, during germination and the addition of Glu had no significant (*p* > 0.05) effect on pterin content. The highest content of endogenous pterin was found in the seedlings treated with Phe and the optimized culture solution.

### 2.6. Gene Expression of Folate Biosynthesis

The expression of four genes—namely, guanosine triphosphate cyclohydrolase I (*GTPCHI*), aminodeoxychorismate synthase (*ADCS*), 6-hydroxymethyldihydropterin pyrophosphokinase/dihydropteroate synthase (*HPPK/DHPS*), and folylpolyglutamate synthetase (*FPGS*)—related to folate biosynthesis were determined. The expression of each gene in the control seedlings was defined as 1.0, and expressions of genes in other seedlings were calculated by comparing them with control seedlings (Figure 4). The lowest expressions of *GTPCHI* (Figure 4A), *HPPK/DHPS* (Figure 4C), and *FPGS* (Figure 4D) in all samples were observed in control seedlings and the seedlings treated with the optimized culture condition had the highest expression of *GTPCH1* (ca. 2.0), *ADCS* (ca. 1.4), and *FPGS* (ca. 2.1). The highest expression of *HPPK/DHPS* was found in *p*ABA treated seedlings (ca. 1.6), and the lowest expression of *ADCS* was observed in both *p*ABA and Glu treated seedlings, which ranged from 0.6 to 0.7. The expressions of *GTPCHI* (Figure 4A) and *ADCS* (Figure 4B) in the Phe treated seedlings were at the same level as that of the seedlings treated with the optimized solution.

## 3. Discussion

Folates play a pivotal role in the primary metabolism of nearly all organisms by functioning as C1-donors or acceptors, and their synthesis in plants can be divided into three steps: pterin synthesis, *p*ABA synthesis, and assembly of these two moieties [14]. As shown in Figure 5, the biosynthesis of pterin branch is conducted in the cytosol, which is initiated by *GTPCHI* and forms 6-hydroxymethyldihydropterin, while *p*ABA is synthesized from chorismite in plastids, which is initiated by *ADCS* [10]. Pterin and *p*ABA moieties are condensed in mitochondria, which is initiated by *HPPK/DHPS* and the polyglutamylation of the resulting folate catalyzed by *FPGS* is the final step [15].

In the present study, a significant positive correlation (*p* < 0.05) was found between the content of total folates with the expression of *GTPCHI*, *ADCS*, and *FPGS* (Appendix A). As the precursors of folate synthesis, *p*ABA was shown to down-regulate the expression of *ADCS* (Figure 4B), which led to the inhibition on the synthesis of endogenous *p*ABA, but up-regulated the *GTPCHI* (Figure 4A), which resulted in the higher production of pterin (Figure 3B). The addition of exogenous Phe and Try promoted the synthesis of *p*ABA (Figure 3A) through up-regulating *ADCS* expression (Figure 4B). This may be due to the negative feedback on the synthesis of endogenous aromatic amino acids, which resulted in the converting of chorismate to *p*ABA, which has also been observed in a study of spinach cultivation [11]. Interestingly, the addition of Phe significantly (*p* < 0.05) increased the content of Tyr (Table 3). Phenylalanine hydroxylase plays a key role in the conversion of Phe to Tyr and tetrahydrobiopterin is the obligatory cofactor of this reaction [17]. This can explain the promotion on pterin synthesis of Phe addition. There was no significant positive correlation (*p* > 0.05) between folate content with the expression of *HPPK/DHPS*, which suggested that condensation of pterin and *p*ABA was not the rate-limiting step during the folate biosynthesis.

Glu is required in the synthesis of folate polyglutamates, and the polyglutamate tail is responsible for the intracellular retention of folate derivatives and binding of folate substrates to enzymes involved in one-carbon transfer or the interconversion of folate derivatives [18]. The addition of Glu had no effect on the expressions of *GTPCHI*, *ADCS*, and *HPPK/DHPS* (Figure 4A–C), which suggested that it cannot influence the synthesis of pterin, *p*ABA, or their condensation in the seedlings. However, a significant increase of *FPGS* expression (Figure 4D) suggested that Glu increased the folate content in wheat seedlings, which was mainly caused by its effect on promoting the rate of polyglutamylation. Considering the mechanism of effects on folate accumulation among aromatic amino acids, Glu, and *p*ABA was different, the combination of these amino acids may have a better promotion effect, which was confirmed through an orthogonal experiment. Moreover, an optimized culture solution containing 1.5 mM Phe, 0.5 mM Tyr, 0.5 mM Try, 0.75 mM *p*ABA, and 0.5 mM Glu was obtained, and it can increase the content of total folates by 50% when compared with the control seedlings.

Folate is a generic term that encompasses the derivatives of this vitamin, distinguished by their highly variable glutamate tail and attachment of one-carbon (C1) units on the pteridine (N5) or the *p*ABA (N10) moiety [19]. In nature, THF is the most reduced form and 10-CHO-THF (formylated on the nitrogen N−10), 5-CHO-THF (formylated on the nitrogen N−5), as well as 5-CH_3_-THF (methylated on the nitrogen 5) are the forms that are commonly found in plant-based foods [20]. It was known that 5-CHO-THF and 5-CH_3_-THF were the main forms of folate in wheat grains [21]. In the present study, it was found that in wheat seedlings, THF and 5-CH_3_-THF were the two most important forms of folate (Figure 2), which together consisted of more than 70% of the total folate content. As the physiologically active form of folate, 5-CH_3_-THF has many advantages, such as low potential for masking the haematological symptoms of vitamin B12 deficiency and inactivating the drugs that inhibit dihydrofolate reductase [22]. Furthermore, 5-CH_3_-THF is more stable than THF during storage and food processing [20]. The RDI of folate is 400 μg for an adult, and 100 g of dry wheat seedlings under red light radiation with the treatments of optimized amino acid solution obtained in this study contained ca. 417 µg of folates, and about 50% of them were 5-CH_3_-THF. Therefore, the bioaccumulation of folates in wheat seedlings through radiation and the addition of amino acids is an economical and effective way to provide natural folates to consumers.

## 4. Materials and Methods

### 4.1. Materials and Chemicals

Wheat grains (*cv.* Huaimai 33) were harvested in 2019 and stored at −20 °C before use. Methanol and acetonitrile (LC-MS grade) were purchased from Fisher Scientific (Gail, Belgium). Rat serum was purchased from Soleibao Technology Co., Ltd. (Beijing, China), and stored in a refrigerator at −80 °C before use. All the other chemicals were analytical grade reagents and purchased from Shoude Biotechnology Co., Ltd. (Nanjing, China).

### 4.2. Germination Conditions

After rinsing with distilled water, wheat grains were disinfected with 1% (*v/v*) sodium hypochlorite for 15 min and soaked in distilled water at 25 °C for 6 h after rinsing with distilled water. Then, the seeds were placed evenly on the seedling tray and germinated in the incubator (LB−300-II, Longyue Instrument Equipment Co., Ltd., Shanghai, China) at 25 °C. The seeds were germinated for 2 days in the dark at 25 °C. From day 3 to day 6, the seedlings were cultivated at 25 °C under the treatment of LED red light (peak at 655 nm with an intensity of 30 μmol/m^2^·s) with a light/dark regime of 16/8 h every day. During germination, the grains were sprayed with different culture solutions for 1 min once an hour and culture solutions were changed every 24 h. After 6 days of germination, seedlings were freeze-dried and stored at −20 °C.

For the single factor test, culture solution treatments were as follows: A. Phenylalanine (Phe) concentrations were 0, 1, 2, 4, and 6 mmol/L; B. Tyrosine (Tyr) concentrations were 0, 0.25, 0.50, 1.0, and 2.0 mmol/L; C. Tryptophan (Trp) concentrations were 0, 0.5, 1.0, 2.0, and 4.0 mmol/L; D. *p*-aminobenzoic acid (*p*ABA) concentrations were 0, 0.25, 0.50, 1.0, and 2.0 mmol/L; E. Glutamic acid (Glu) concentrations were 0, 0.50, 1.0, 1.5, and 2.0 mmol/L; F. γ-aminobutyric acid (GABA) concentrations were 0, 1, 2, 4, and 6 mmol/L. Based on single factor test results, an orthogonal experiment was designed to include five factors (Phe, Tyr, Trp, *p*ABA, and Glu) and four levels to optimize the amino acid culture solutions for better accumulation of folates in wheat seedlings.

### 4.3. Determination of Folates, pABA and Pterin

Extraction and quantification of folates were conducted following the method of Riaz et al. [21] with some modifications. The freeze-dried seedlings were milled and extracted with 5 mM sodium phosphate buffer solution (pH 7.2), which contained 1% sodium ascorbate and 0.2% β-mercaptoethanol, for 10 min under a boiling water bath. After cooling in ice and centrifugation (13,000× *g*, 10 min), supernatants were mixed with rat serum and incubated at 37 °C for 4 h to deconjugate the polyglutamylated tails. After that, mixtures were boiled for 10 min and centrifugated (13,000× *g*, 10 min) at 4 °C. The supernatants were ultra-filtrated at 4 °C for 20 min on a 3 kDa ultra-filtration tube (Millipore, Burlington, MA, USA) before analysis.

The extraction and quantification of *p*ABA was conducted following the method of Orsomando, et al. [23]. Briefly, tissues were pulverized in liquid N_2_ and homogenized with 7 mL of methanol. The methanol extracts were treated with nitrogen flow and the residues were treated with ultrasound for 5 min after being dissolved in the distilled water. The samples were then mixed with 2 M HCl and the supernatants from the centrifugation (12,000× *g*, 30 min) were used for the analysis. The extraction and quantification of pterin were performed according to Ramírez Rivera, et al. [24] with some modification. Briefly, tissues were mixed with an extract solution, which contained methanol, chloroform, and water in a ratio of 12:5:1 (*v/v/v*) and incubated at 50 °C for 5 min. After that, mixtures were added with 0.5 mL chloroform and 0.75 mL water. The aqueous phase of the extract was freeze-dried and oxidized by reacting with 1 M HCl containing 1% I_2_ and 2% KI in darkness for 1 h before analysis.

A method of high-performance liquid chromatography (HPLC) followed by tandem mass spectrometry (MS/MS) was employed for identification and quantification. Chromatographic analyses were performed on an SCIEX ExionLC™ HPLC system (Foster City, CA, USA) with an Akzo Nobel analytical column (Kinetex, 2.6 µm F5 C18, 50 × 4.6 mm). For folate analysis, the injection volume was 2.0 μL with a flow rate of 0.30 mL/min. The temperatures of the injector and column oven were 4 °C and 30 °C, respectively. The mobile phase involved water (mobile phase A) and acetonitrile (mobile phase B), both of which contained 0.1% (*v/v*) formic acid. The proportion of mobile phase B increased linearly from 2% to 98% from 0 min to 3 min and stayed at 98% for 1 min. Then, the proportion of mobile phase B decreased to 2% in 0.1 min, followed by a subsequent equilibration. For determination, the mobile phases were water (contained 0.1% formic acid) and methanol (contained 0.1% formic acid) and were eluted isocratically at 0.6 mL/min.

An SCIEX Triple Quad™ 5500 triple quadruple tandem MS coupled with an ESI (electron spray ionization) interface was used for the analysis of five folate analogues: 5-methyltetrahydrofolate (5-CH_3_-THF), 10-formylfolic acid (10-CHO-FA), folic acid (FA), tetrahydrofolate (THF) and 5-formyltetrahydrofolate (5-CHO-THF), *p*ABA, and pterin. The source temperature was adjusted to 550 °C and the ion spray voltage to 5500 V. Nitrogen was used as gas 1 (55 psig), gas 2 (55 psig), curtain gas (35 psig), and collision-activated dissociation gas (8 psig). Compound parameters are given in Appendix A.

### 4.4. Determination of Free Amino Acids

The contents of free amino acids were determined according to the method of Yoon, et al. [25] with minor modifications. Freeze-dried wheat seedlings (0.5 g) were mixed with a 10 mL trichloroacetic acid solution (5% *v/v*) and incubated for 1 h at 24 °C. After that, the mixtures were centrifuged (3000× *g*, 15 min) and the supernatants were filtered through a syringe filter (0.45 μm). The contents of the free amino acids were measured by an amino acid analyzer (L−8800, Hitachi, Tokyo, Japan).

### 4.5. Expression of Genes

Frozen wheat seedlings were ground, and their RNA were extracted by an RNA extraction kit (catalog no. 9769, Takara, Shiga, Japan). The first strand of cDNA was synthesized using the quantitative real-time polymerase chain reaction (RT-PCR) Master Mix Kit (Takara, catalog no. RR036A). Quantification analysis was conducted by the SYBR Premix ExTaqTM (Takara, catalog no. RR420A) with an ABI 7500 PCR system (Applied Biosystems, Foster City, CA, USA). The primers of genes encoding the four essential enzymes related to folate biosynthesis are listed in Appendix A.

### 4.6. Statistical Analysis

The results were presented as mean and standard deviations of three replicates. Statistical analysis and Pearson correlation between production of folates and expression of four enzymes were performed by One-way analysis of variance (ANOVA) by SPSS 19.0 (SPSS Inc., Chicago, IL, USA) at a *p* < 0.05.

## 5. Conclusions

During germination of wheat seedlings, the addition of phenylalanine and tyrosine enhanced the folate accumulation by up-regulating the expression of *GTPCHI* and *ADCS* while *p*-aminobenzoic acid and glutamic acid enhanced the accumulation by up-regulating the expression of *HPPK/DHPS* and *FPGS*, respectively. A combination of amino acids (1.5 mM phenylalanine, 0.5 mM tyrosine, 0.5 mM tryptophan, 0.75 mM *p*-aminobenzoic acid, and 0.5 mM glutamic acid) was obtained through the orthogonal design, which can increase the folate accumulation by about 50% during germination under red light radiation. The present study proved that the germination of wheat seedlings with radiation and the addition of amino acids is a promising method for providing natural folates to consumers.

## Figures and Tables

**Figure 1 molecules-27-06868-f001:**
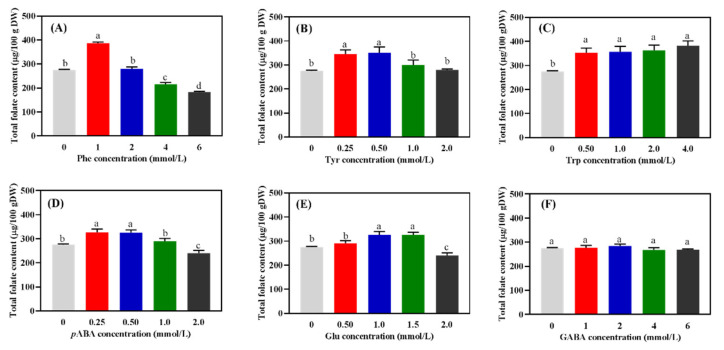
The content of total folates in wheat seedlings with different treatments (*n* = 3). Phe (**A**), Tyr (**B**), Trp (**C**), *p*ABA (**D**), Glu (**E**), and GABA (**F**) were seedlings treated with phenylalanine, tyrosine, tryptophan, *p*-aminobenzoic acid, γ-aminobutyric acid, and glutamate, respectively. ^a–d^ represent significant differences among different treatment concentrations (*p* < 0.05).

**Figure 2 molecules-27-06868-f002:**
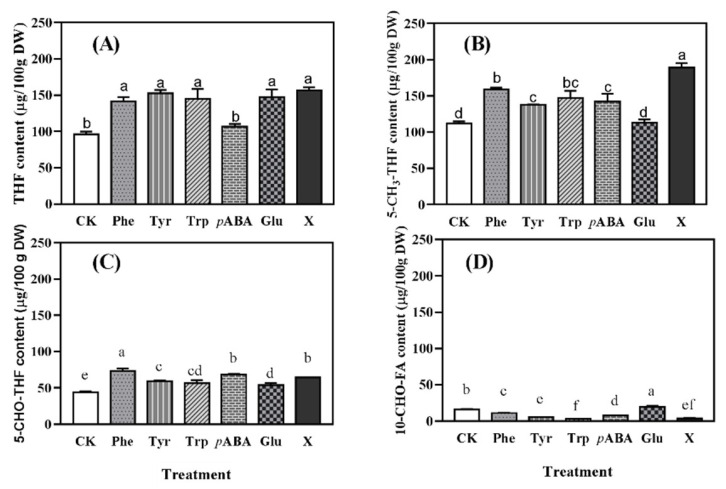
Contents of tetrahydrofolate (**A**: THF), 5-methyltetrahydrofolate (**B**: 5-CH_3_-THF), 5-formyltetrahydrofolate (**C**: 5-CHO-THF), and 10-formylfolic acid (**D**: 10-CHO-FA) in wheat seedlings. CK was the wheat seedlings cultured with distilled water; Phe, Tyr, Trp, *p*ABA, and Glu were seedlings cultured with 1.00 mM phenylalanine, 0.25 mM tyrosine, 0.50 mM tryptophan, 0.25 mM *p*-aminobenzoic acid, and 1.00 mM glutamate, respectively; X was seedlings cultured with the optimized culture solution (1.50 mM phenylalanine, 0.50 mM tyrosine, 0.50 mM tryptophan, 0.75 mM *p*-aminobenzoic acid, and 0.50 mM glutamate). ^a–f^ represent significant differences among treatment factors (*p* < 0.05).

**Figure 3 molecules-27-06868-f003:**
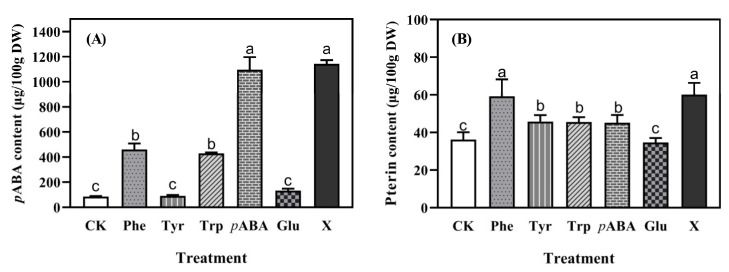
Contents of *p*-aminobenzoic acid (*p*ABA); (**A**) and pterin (**B**) in wheat seedlings. CK was wheat seedlings cultured with distilled water; Phe, Tyr, Trp, *p*ABA, and Glu were seedlings cultured with 1.00 mM phenylalanine, 0.25 mM tyrosine, 0.50 mM tryptophan, 0.25 mM *p*-aminobenzoic acid, and 1.00 mM glutamate, respectively; X was seedlings cultured with the optimized culture solution (1.50 mM phenylalanine, 0.50 mM tyrosine, 0.50 mM tryptophan, 0.75 mM *p*-aminobenzoic acid, and 0.50 mM glutamate). ^a–c^ represent significant differences among treatment factors (*p* < 0.05).

**Figure 4 molecules-27-06868-f004:**
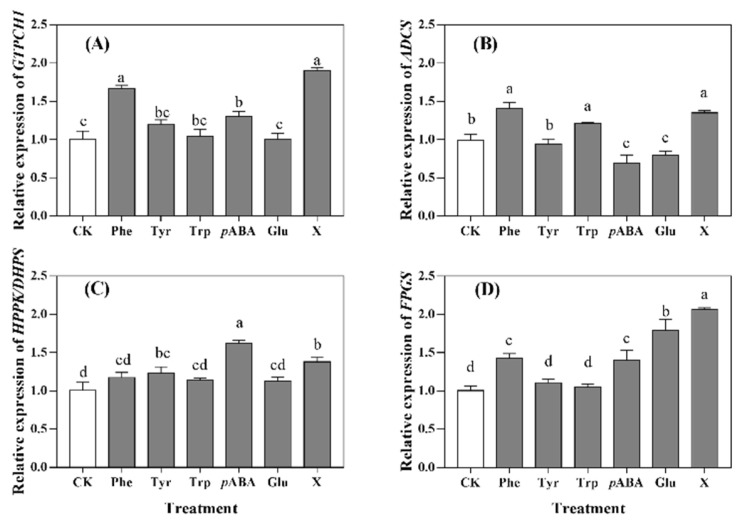
Relative expression of *GTPCHI* (**A**), *ADCS* (**B**), *HPPK*/*DHPS* (**C**) and *FPGS* (**D**) under treatments of different amino acid. CK was wheat seedlings cultured with distilled water; Phe, Tyr, Trp, *p*ABA, and Glu were seedlings cultured with 1.00 mM phenylalanine, 0.25 mM tyrosine, 0.50 mM tryptophan, 0.25 mM *p*-aminobenzoic acid, and 1.00 mM glutamate, respectively; X was seedlings cultured with the optimized culture solution (1.50 mM phenylalanine, 0.50 mM tyrosine, 0.50 mM tryptophan, 0.75 mM *p*-aminobenzoic acid, and 0.50 mM glutamate). ^a–d^ represent significant differences among treatment factors (*p* < 0.05).

**Figure 5 molecules-27-06868-f005:**
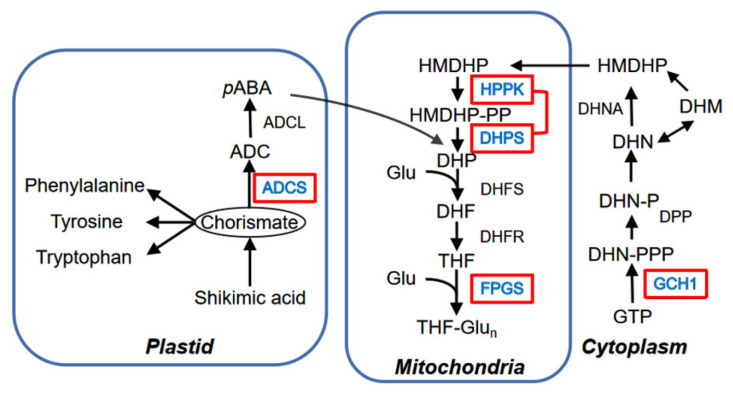
Schematic representation of the folate biosynthesis pathway in plants. Chemical compounds: ADC: Aminodeoxychorismate; DHF: Dihydrofolate; DHM: Dihydromyricetin; DHN: Dihydroneopterin; DHP: Dihydropteroate; Glu: Glutamine; GTP: Guanosine−5′-triphosphate; HMDHP: 6-Hydroxymethyldihydropterin; -P/-PP/-PPP: mono/di/triphosphate; *ρ*-ABA: Para-aminobenzoate; THF: Tetrahydrofolate; THF-Glu*_n_*: Tetrahydrofolate polyglutamate; Enzymes: ADCL: aminodeoxychorismate lyase; ADCS: aminodeoxychorismate synthase; DHFS: Dihydrofolate synthase; DHFR: Dihydrofolate synthase reductase; DHNA: Dihydroneopterin aldolase FPGS: Folylpolyglutamate synthetase; GCH1: GTP cyclohydrolase I; HPPK/DHPS: Pyrophosphokinase/dihydropteroate synthase [16].

**Table 1 molecules-27-06868-t001:** Orthogonal experiment on the effect of amino acids on folate accumulation during wheat germination (*n* = 3).

Number	Factors					Total Folate Content (µg/100 g DW)
	A Phe (mmol/L)	B Tyr (mmol/L)	C Trp (mmol/L)	D *p*ABA (mmol/L)	E Glu (mmol/L)	
1	1 (0)	1 (0)	1 (0)	1 (0)	1 (0.5)	278.20 ± 6.54
2	1	2 (0.25)	2 (0.25)	2 (0.25)	2 (1.0)	321.18 ± 20.25
3	1	3 (0.50)	3 (0.50)	3 (0.50)	3 (1.5)	300.75 ± 1.82
4	1	4 (0.75)	4 (0.75)	4 (0.75)	4 (2.0)	321.56 ±17.33
5	2 (0.50)	1	2	3	4	316.45 ± 27.66
6	2	2	1	4	3	328.81 ± 16.82
7	2	3	4	1	2	344.93 ± 9.73
8	2	4	3	2	1	364.31 ± 19.21
9	3 (1.0)	1	3	4	2	319.96 ± 17.84
10	3	2	1	3	1	351.35 ± 14.14
11	3	3	4	2	4	308.98 ± 12.01
12	3	4	2	1	3	296.55 ± 3.90
13	4 (1.5)	1	4	2	3	326.80 ± 16.59
14	4	2	3	1	4	345.00 ± 22.84
15	4	3	2	4	1	356.92 ± 1.48
16	4	4	1	3	2	342.49 ± 1.20
K_1_	305.43	310.35	325.22	316.18	343.31	
K_2_	338.62	336.59	328.40	330.32	332.14	
K_3_	319.22	333.51	332.50	327.77	313.23	
K_4_	348.42	331.23	325.57	337.44	323.00	
Best level	A4	B2	C3	D4	E1	
R	42.99	26.24	7.28	21.26	30.08	
Ranking	A > E > B > D > C					

Phe, Tyr, Trp, *p*ABA, and Glu were seedlings treated with phenylalanine, tyrosine, tryptophan, *p*-aminobenzoic acid, and glutamate, respectively.

**Table 2 molecules-27-06868-t002:** Arrangement and result of validation trials (*n* = 3).

	Phe (mmol/L)	Tyr (mmol/L)	Trp (mmol/L)	*p*ABA (mmol/L)	Glu (mmol/L)	Total Folate Content (µg/100 g DW)
Optimal combination	1.5	0.50	0.50	0.75	0.5	417.24 ± 7.14 ^a^
Random combination 1	1.5	0.75	0.25	0.75	0.5	349.26 ± 17.98 ^b^
Random combination 2	0.5	0	0.25	0.50	2.0	328.59 ± 15.39 ^b^
Random combination 3	1.0	0.50	0.75	0.25	2.0	306.83 ± 1.13 ^c^

Phe, Tyr, Trp, *p*ABA, and Glu indicate phenylalanine, tyrosine, tryptophan, *p*-aminobenzoic acid, and glutamate, respectively. Data are expressed as the mean ± SD (*n* = 3). ^a–c^ represent significant difference among treatment factors (*p* < 0.05).

**Table 3 molecules-27-06868-t003:** Concentrations (mg/100 g DW) of free amino acids in wheat seedlings cultured with amino acids (*n* = 3).

Treatment							
	CK	Phe	Tyr	Trp	*p*ABA	Glu	X
Asparagine	31.66 ± 0.47 ^d^	40.10 ± 0.01 ^b^	36.86 ± 0.04 ^c^	37.32 ± 0.30 ^c^	32.80 ± 0.21 ^d^	44.02 ± 0.71 ^a^	36.11 ± 0.20 ^c^
Threonine	151.49 ± 1.23 ^a^	159.14 ± 2.09 ^a^	142.90 ± 1.59 ^a^	137.00 ± 1.62 ^a^	97.45 ± 0.27 ^b^	159.00 ± 3.57 ^a^	96.97 ± 0.13 ^b^
Serine	-*	-	-	-	-	-	-
Glutamine	100.26 ± 0.13 ^f^	120.87 ± 0.54 ^b^	101.14 ± 0.11 ^f^	112.17 ± 0.35 ^d^	108.54 ± 0.49 ^e^	116.53 ± 1.16 ^c^	126.41 ± 0.63 ^a^
Glycine	11.45 ± 0.05 ^b^	11.38 ± 0.03 ^b^	11.96 ± 0.02 ^a^	11.29 ± 0.04 ^bc^	11.11 ± 0.02 ^c^	11.88 ± 0.13 ^a^	10.45 ± 0.04 ^d^
Alanine	36.44 ± 0.03 ^f^	44.54 ± 0.09 ^a^	40.84 ± 0.12 ^d^	42.04 ± 0.20 ^c^	35.98 ± 0.06 ^f^	43.35 ± 0.29 ^b^	38.82 ± 0.10 ^e^
Cysteine	19.24 ± 0.11 ^a^	18.79 ± 0.10 ^b^	18.68 ± 0.02 ^bc^	16.77 ± 0.07 ^e^	18.49 ± 0.07 ^cd^	18.73 ± 0.03 ^bc^	18.31 ± 0.03 ^d^
Valline	86.29 ± 0.09 ^b^	91.03 ± 0.31 ^a^	77.83 ± 0.37 ^e^	0.74 ± 0.01 ^f^	82.11 ± 0.12 ^c^	81.34 ± 0.50 ^cd^	80.77 ± 0.13 ^d^
Methionine	14.27 ± 0.05 ^a^	13.34 ± 0.11 ^b^	12.80 ± 0.02 ^cd^	12.64 ± 0.06 ^d^	14.36 ± 0.03 ^a^	13.02 ± 0.14 ^c^	14.11 ± 0.07 ^a^
Isoleucine	77.60 ± 0.08 ^b^	79.38 ± 0.04 ^a^	69.50 ± 0.32 ^f^	67.97 ± 0.11 ^g^	74.54 ± 0.01 ^d^	70.30 ± 0.44 ^e^	76.11 ± 0.05 ^c^
Leucine	78.85 ± 0.05 ^c^	81.53 ± 0.22 ^a^	69.84 ± 0.24 ^e^	68.30 ± 0.14 ^g^	75.44 ± 0.06 ^d^	69.00 ± 0.36 ^f^	80.32 ± 0.11 ^b^
Tyrosine	48.86 ± 0.25 ^d^	70.04 ± 0.10 ^b^	48.87 ± 0.17 ^d^	46.38 ± 0.13 ^e^	49.84 ± 0.01 ^c^	44.77 ± 0.27 ^f^	71.27 ± 0.09 ^a^
Phenylalanine	85.10 ± 0.13 ^d^	468.98 ± 2.30 ^a^	79.40 ± 0.14 ^e^	90.35 ± 0.17 ^c^	90.17 ± 0.15 ^c^	86.94 ± 0.22 ^d^	162.68 ± 0.28 ^b^
Lysine	51.09 ± 0.23 ^c^	49.25 ± 0.11 ^d^	49.79 ± 0.40 ^d^	49.50 ± 0.04 ^d^	51.77 ± 0.14 ^c^	53.36 ± 0.20 ^b^	54.36 ± 0.10 ^a^
Histidine	60.40 ± 0.39 ^a^	60.62 ± 0.26 ^a^	57.31 ± 0.13 ^b^	51.66 ± 0.07 ^d^	61.04 ± 0.21 ^a^	56.51 ± 0.16 ^c^	55.89 ± 0.02 ^c^
Argnine	56.81 ± 0.23 ^c^	53.46 ± 0.02 ^e^	54.58 ± 0.23 ^d^	54.60 ± 0.12 ^d^	60.12 ± 0.26 ^b^	61.69 ± 0.22 ^a^	62.13 ± 0.07 ^a^
Proline	67.97 ± 0.12 ^c^	64.32 ± 0.02 ^d^	55.40 ± 0.46 ^f^	64.36 ± 0.25 ^d^	74.99 ± 0.34 ^b^	61.70 ± 0.35 ^e^	82.83 ± 0.24 ^a^
Total	977.77 ± 1.29 ^c^	1426.75 ± 1.54 ^a^	927.70 ± 4.13 ^d^	936.25 ± 0.68 ^d^	910.97 ± 26.08 ^d^	992.16 ± 8.45 ^c^	1067.55 ± 0.66 ^b^

CK was the wheat seedlings cultured with distilled water; Phe, Tyr, Trp, pABA, and Glu were seedlings cultured with 1.00 mM phenylalanine, 0.25 mM tyrosine, 0.50 mM tryptophan, 0.25 mM, *p*-aminobenzoic acid, and 1.00 mM glutamate, respectively; X was the seedlings cultured with the optimized culture solution (1.50 mM phenylalanine, 0.50 mM tyrosine, 0.50 mM tryptophan, 0.75 mM *p*-aminobenzoic acid, and 0.50 mM glutamate). ^a–f^ represent significant differences among treatment factors (*p* < 0.05). Data are expressed as the mean ± SD. * Not detected.

## Data Availability

The data presented in this study are available on request from the corresponding author.

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
