# Peer review of "Effect of Amino Acids on Folates Accumulation in Wheat Seedlings during Germination under Red Light Radiation"

_molecules, 2022, doi:10.3390/molecules27206868_

Round 1

Reviewer 1 Report

Comments to Author In this manuscript the authors investigated the effect of amino acids on folates accumulation in wheat seedling during germination under red light radiation, along with the expression of related genes. They concluded that some amino acids can  enhance the accumulation of folates in wheat seedlings and established a optimized culture solution for the folates enrichment. The manuscript is interesting and can be accepted for publication after some minor revisions.  1. Page 1 in line 33 change the reference ” (Krupenko & Horita, 2019)” to the correct form.   2. Page 1 in line 43 “Triticum aestivum” should be italicized. 3. Page 7 in lines 211-213:“The up-regulate of ADCS expression was also observed which can confirm the increasing of pABA synthesis in the wheat seedlings treated with Phe, Tyr, or Try”. The meaning of this sentence is not clear, please modify. 4. Page 8 in line 262: change “From day 2 to day 6” to “From day 3 to day 6” 5. In footnotes of table and figures, change “(1mM phenylalanine 0.25 mM Tyrosine 0.50 mM Tryptophan 0.25 mM p-aminobenzoic acid, and 1.0 mM Glutamate).” to “(1mM phenylalanine, 0.25 mM Tyrosine, 0.50 mM Tryptophan, 0.25 mM p-aminobenzoic acid, and 1.0 mM Glutamate).”

Author Response

  1. Page 1 in line 33 change the reference ” (Krupenko & Horita, 2019)” to the correct form.   

Answer: Thanks for the suggestion. The references have been modified (Page 1 Line 33).

  1. Page 1 in line 43 “Triticum aestivum” should be italicized. 

Answer: The term has been italicized (Page 1 Line 43).

  1. Page 7 in lines 211-213:“The up-regulate of ADCS expression was also observed which can confirm the increasing of pABA synthesis in the wheat seedlings treated with Phe, Tyr, or Try”. The meaning of this sentence is not clear, please modify.

Answer: The sentence has been modified.

  1. Page 8 in line 262: change “From day 2 to day 6” to “From day 3 to day 6” 5. In footnotes of table and figures, change “(1mM phenylalanine 0.25 mM Tyrosine 0.50 mM Tryptophan 0.25 mM p-aminobenzoic acid, and 1.0 mM Glutamate).” to “(1mM phenylalanine, 0.25 mM Tyrosine, 0.50 mM Tryptophan, 0.25 mM p-aminobenzoic acid, and 1.0 mM Glutamate).”

Answer: The footnotes have been modified.

Reviewer 2 Report

Page 1. Line 33. Please standardize the references as described in author guideline

 Page 1, Line 43. Scientific name, double check, pl

 Page 2, Line 50-51. Glutamic acid (Glu) and pABA are precursors in folate biosynthesis and addition of 50 these molecules have been shown to enhance folate synthesis in plants (Zhao, et al., 21). I suggest that the author reread de Zhao, et al because these author write: “Since phenolics are not related to GABA metabolism, it is speculated that GABA acts as a signal molecule to mediate the synthesis of phenolics in soybean sprouts” Moreover Zhao is cited in discordance of authors guide.

 There are any confuse in this Page 2, figure 1. This figure did not in 300 DPI. I suggest a better presentation of this figure, maybe using colors

 Page 2, Line 72. I suggest that the authors redo the statistical analyses, because a 32% increase in folates in seedlings cannot be non-significant. If not, check if the values follow the normal distribution trend, paradigm of parametric means tests.

 I suggest to the authors that I presented the figure and the part of the figure where the data that are presented are, type figure 1A, figure 2D, citing only the figure makes it a little difficult to read

 Page 3, lines 85-87. “To optimize the combination of amino acids for folate accumulation in the wheat seedling, an orthogonal design was employed and four levels of five factors (Phe, Tyr, 86 Try, pABA, and Glu) were selected based on the result of the single -factor experiments.” What do the authors call single-factor-experiments?

 Page 3, lines 89-93. “By comparing different levels of K values, it was estimated that the best amino acid culture solution combination for accumulation of folate was A4B2C3D4E1, which contained 1.5 mM 91 Phe, 0.25 mM Tyr, 0.50 mM Trp, 0.75 mM pABA and 0.5 mM Glu.” This text seems confusing to me. I ask for a better essay

 Page 4, lines 103-105. The text and figures call amino acids by their world abbreviations. As much as they are well known, they should be presented the first time they are mentioned, not only in the results, such as footnote.

 Page 4, figure 2. The y axis of all figure may have the same strength because with different values of y, they confuse the reader with a false impression of high values

 Figures and tables should be self-explanatory, to the point that I don't have to go back to the text to understand them. So they cannot have abbreviations that are not presented in them, even if this is done in 20 tables, or 20 figures.

 I understand each one as an individual figure Page 4-5, lines 129-142, the text is very boring to read, the authors must improve this writing. It is complicated to understand the text and the collocations overlap making the text even more tiring. I was expecting a more metabolic explanation than saying it increased or decreased. I remind the authors that the your chose is Molecules

 Section 2.6. Genes are written in italics, double check, pl

 The data presented in section 2.6 could be better presented in the form of a heatmap, this would be very interesting and avoid different y-axis values, which at first sight confuse the reader First paragraph of the discussion is very theoretical and boring to read, I would expect a metabolic map or an explanatory figure than a poorly written text.

 Page 7, Lines 207-209 presents a sentence with no meaning and no causality in context. This is public knowledge and doesn't change anything at work, I suggest deleting it.

 Page 7, lines 209-211 would expect a heatmap or correlation analysis here. The presentation of this item is very weak. Furthermore, this is a result, not a discussion.

 The study doesn’t present a discussion section, instead a restatement of the results in other words. This item has to be completely rewritten and within the profile of Molecules articles

 Page 9, line 280: 5 mM phosphate buffer. Which of the phosphate buffers was used?

 Page 9, line 280: As written it looks like the pH was adjusted after adding 1% sodium ascorbate and 0.2% β-mercaptoethanol. Was it that or was the buffer already 7.2 before the addition of these compounds? I was in doubt

 Page 9, line 282, What is the role of rat serum in the reaction?

 Page 9, lines 209-210. buffer (20 mM 289 ammonium acetate, what is the pH?

 A 20 mM ammonium acetate was mixed with 2M HCl, what is the purpose of the buffer in this reaction? 2 M HCl destroys any buffer that is intended to be used

 The conclusions (last paragraph of discussion) are very weak

Author Response

Page 1. Line 33. Please standardize the references as described in author guideline

Answer: Thanks for the suggestion. The reference have been modified.

Page 1, Line 43. Scientific name, double check, pl

Answer: The Scientific name has been changed in the italic font (Page 1 Line 43).

Page 2, Line 50-51. Glutamic acid (Glu) and pABA are precursors in folate biosynthesis and addition of 50 these molecules have been shown to enhance folate synthesis in plants (Zhao, et al., 21). I suggest that the author reread de Zhao, et al because these author write: “Since phenolics are not related to GABA metabolism, it is speculated that GABA acts as a signal molecule to mediate the synthesis of phenolics in soybean sprouts” Moreover Zhao is cited in discordance of authors guide.

Answer: The sentence has been changed to “Gamma-aminobutyric acid (GABA) is a non-protein amino acid which plays a versatile role in plants and can mediate the synthesis of many bioactive compounds, such as phenolics, as a signal molecule” (Page 2 Lines 54-55).

There are any confuse in this Page 2, figure 1. This figure did not in 300 DPI. I suggest a better presentation of this figure, maybe using colors

Answer: Thanks for the suggestion. The figure 1 has been changed to 300 DPI with colors.

Page 2, Line 72. I suggest that the authors redo the statistical analyses, because a 32% increase in folates in seedlings cannot be non-significant. If not, check if the values follow the normal distribution trend, paradigm of parametric means tests.

Answer: Sorry for the misleading. We want to present that there was no significant difference among contents of folates in the sprouts treated 0.5, 1.0, 2.0 or 4.0 mmol/L Trp. But the folates in the sprouts treated with Trp had at least 32% increase of folates content compared to no addition of Trp. We have modified the sentence to “Comparing with the sprout without any addition, application of Trp (0.5 mM) increased the content of folates in the wheat seedlings by ca. 32% (Figure 1C). However, increasing the Trp addition (0.5-4.0 mM) cannot enhance its promotion on the folate accumulation (Figure 1C).” In Page 2 Lines 74-77).

I suggest to the authors that I presented the figure and the part of the figure where the data that are presented are, type figure 1A, figure 2D, citing only the figure makes it a little difficult to read

Answer: Thanks for the suggestion, we have modified some sentences in Results and Discussion to better present the results.

 Page 3, lines 85-87. “To optimize the combination of amino acids for folate accumulation in the wheat seedling, an orthogonal design was employed and four levels of five factors (Phe, Tyr, 86 Try, pABA, and Glu) were selected based on the result of the single -factor experiments.” What do the authors call single-factor-experiments?

Answer: The sentence has been changed to “results about the effects of different levels of individual amino acid on the folates accumulation (Result 2.1)”. In Page 3 Lines 91-92.

 Page 3, lines 89-93. “By comparing different levels of K values, it was estimated that the best amino acid culture solution combination for accumulation of folate was A4B2C3D4E1, which contained 1.5 mM 91 Phe, 0.25 mM Tyr, 0.50 mM Trp, 0.75 mM pABA and 0.5 mM Glu.” This text seems confusing to me. I ask for a better essay

Answer: Orthogonal experimental design is a scientific test design method for selecting the right amount of representative points or using cases from a large number of experimental data, so as to arrange experiments or tests reasonably. K values were calculated based on data from the experiments and can be used for predicting the best combination of factors for accumulating folates in this study.

In each factor, the higher K value indicate the higher production of folates. Therefore, the best production will be achieved, theoretically, when sprout was treated with the combination of the highest K value from each factors (that is A4B2C3D4E1). A4B2C3D4E1 means the treatment of 1.5 mM Phe, 0.25 mM Tyr, 0.50 mM Trp, 0.75 mM pABA and 0.5 mM Glu.

 Page 4, lines 103-105. The text and figures call amino acids by their world abbreviations. As much as they are well known, they should be presented the first time they are mentioned, not only in the results, such as footnote.

Answer: All the full name of amino acids have been given in all figures or their footnotes.

 Page 4, figure 2. The y axis of all figure may have the same strength because with different values of y, they confuse the reader with a false impression of high values

Answer: The figure 2 has been changed to have the same y axis.

 Figures and tables should be self-explanatory, to the point that I don't have to go back to the text to understand them. So they cannot have abbreviations that are not presented in them, even if this is done in 20 tables, or 20 figures.

Answer: We have carefully checked that explanations were given in the footnotes of figures and tables to make the figures and tables self-explanatory.

 I understand each one as an individual figure Page 4-5, lines 129-142, the text is very boring to read, the authors must improve this writing. It is complicated to understand the text and the collocations overlap making the text even more tiring. I was expecting a more metabolic explanation than saying it increased or decreased. I remind the authors that the your chose is Molecules

Answer: We have tried our best to make the paragraph concise (Page 5, Lines137-148).

 Section 2.6. Genes are written in italics, double check, pl

Answer: Thanks for the suggestion. We have changed the genes into italics.

The data presented in section 2.6 could be better presented in the form of a heatmap, this would be very interesting and avoid different y-axis values, which at first sight confuse the reader

Answer: Thanks for the suggestion. The y-axis values in figure 4 have been unified to avoid confuse. A correlation analysis has been added as Table S1.

 First paragraph of the discussion is very theoretical and boring to read, I would expect a metabolic map or an explanatory figure than a poorly written text.

Answer: A explanatory figure has been added (Figure 5) 

 Page 7, Lines 207-209 presents a sentence with no meaning and no causality in context. This is public knowledge and doesn't change anything at work, I suggest deleting it.

Answer: The sentence has been deleted.

 Page 7, lines 209-211 would expect a heatmap or correlation analysis here. The presentation of this item is very weak. Furthermore, this is a result, not a discussion.

Answer: A correlation analysis about folates production and expression of enzymes has been added as Table S1.

The study doesn’t present a discussion section, instead a restatement of the results in other words. This item has to be completely rewritten and within the profile of Molecules articles

Answer: The discussion section has been modified.

 Page 9, line 280: 5 mM phosphate buffer. Which of the phosphate buffers was used?

Answer: A phosphate buffer solution (PBS) with pH 7.2 was used.

 Page 9, line 280: As written it looks like the pH was adjusted after adding 1% sodium ascorbate and 0.2% β-mercaptoethanol. Was it that or was the buffer already 7.2 before the addition of these compounds? I was in doubt

Answer: The buffer was in pH 7.2 before adding 1% sodium ascorbate and 0.2% β-mercaptoethanol. To avoid misleading, this sentence has been changed to “...extracted with 5 mM phosphate buffer solution (pH 7.2), which contained 1% sodium ascorbate and 0.2% β-mercaptoethanol, for 10 min...”. In page 9, lines 321-322

 Page 9, line 282, What is the role of rat serum in the reaction?

Answer: The rat serum was used for deconjugation of the polyglutamylated tail. The sentence has been changed to “...mixed with rat serum and incubated at 37 °C for 4 h to deconjugate the polyglutamylated tails.” In page 9, lines 324-325

 Page 9, lines 209-210. buffer (20 mM 289 ammonium acetate, what is the pH?

Answer: The pH was 8.0.

 A 20 mM ammonium acetate was mixed with 2M HCl, what is the purpose of the buffer in this reaction? 2 M HCl destroys any buffer that is intended to be used

Answer: There were free form and bound form of pABA in the samples and adding 2 M HCL can release the bound form of pABA.

 The conclusions (last paragraph of discussion) are very weak

Answer: There is a conclusion in section 5 Conclusion. The last paragraph of discussion was present to discuss the transformation of different folates and its meaning in nutrition.

Round 2

Reviewer 2 Report

I repeat what I said in the first round, in this study the authors forgot to add the discussion section because everything that is written in the discussion section is not a real discussion rather than a simple restatement of the results. May have be improved

 One I asked a simple question, and the authors did not know how to answer me. Which phosphate buffer was used, and the response was PBS. PBS is a generic name. Unless I'm wrong and the authors can enlighten me there are basically two types of phosphate buffer: the potassium phosphate buffer and the sodium phosphate buffer. By analogy I believe that the second was used, but the authors should make it clear in the study what type of buffer was used as this is the basis of science, repeatability.

This study presents very confusion sentences and this way it is making it difficult for the work to be repeated by other research groups I ask that the authors be accurate in informing the line where the corrections are found because in this version no correction was presented in the indicated line. I also ask that any correction be presented highlighted in the file in red or with control track.

The authors did not answer the question asked in the first round: "A 20 mM ammonium acetate was mixed with 2M HCl, what is the purpose of the buffer in this reaction? 2 M HCl destroys any buffer that is intended to be used"

The conclusion presented does not support the objectives and data presented, it must be rewritten

Author Response

I repeat what I said in the first round, in this study the authors forgot to add the discussion section because everything that is written in the discussion section is not a real discussion rather than a simple restatement of the results. May have be improved

Response: We have removed some sentences about the results from Discussion. Please check. But it is inevitable to use some restatements of the results for introducing further discussion about the reason behind the changes of some parameters or the meaning of the results from the nutritional point of view. 

One I asked a simple question, and the authors did not know how to answer me. Which phosphate buffer was used, and the response was PBS. PBS is a generic name. Unless I'm wrong and the authors can enlighten me there are basically two types of phosphate buffer: the potassium phosphate buffer and the sodium phosphate buffer. By analogy I believe that the second was used, but the authors should make it clear in the study what type of buffer was used as this is the basis of science, repeatability.

Response: We apologize for the confusion. You are right that we used sodium phosphate buffer. The sentence has been changed to “extracted with 5 mM sodium phosphate buffer solution”

This study presents very confusion sentences and this way it is making it difficult for the work to be repeated by other research groups I ask that the authors be accurate in informing the line where the corrections are found because in this version no correction was presented in the indicated line. I also ask that any correction be presented highlighted in the file in red or with control track.

Response: We uploaded both a version with note of correction and a clean version to the system in the first round of revision. But seems like it only show you the clean version. We will use the version with note of correction from last time and made the correction according to your comments on it. We hope this will be ok.

The authors did not answer the question asked in the first round: "A 20 mM ammonium acetate was mixed with 2M HCl, what is the purpose of the buffer in this reaction? 2 M HCl destroys any buffer that is intended to be used"

Response: After checking the document of experiments, we found it was only water instead of buffer. We are really sorry for making this terrible mistake. We mixed up the method for pABA determination with the other experiment. Thanks a lot for pointing it out. We have modified the sentence to “…5 min after dissolving in the distilled water. The samples were then mixed with 2 M HCl…”

The conclusion presented does not support the objectives and data presented, it must be rewritten

Response: The conclusion has been rewritten to “During germination of wheat seedling, addition of phenylalanine and tyrosine enhanced the folate accumulation by up-regulating the expression of GTPCHI and ADCS while p-aminobenzoic acid and glutamic acid enhanced the accumulation by up-regulating the expression of HPPK/DHPS and FPGS, respectively. A combination of amino acids (1.5 mM phenylalanine, 0.5 mM tyrosine, 0.5 mM tryptophan, 0.75 mM p-aminobenzoic acid and 0.5 mM glutamic acid) was obtained through the orthogonal design, which can increase the folate accumulation by about 50% during germination under red light radiation. The present study proved germination of wheat seedlings with radiation and addition of amino acids is a promising way to provide natural folates to the consumers.”